# Cannabidiol Modifies the Glutamate Over-Release in Brain Tissue of Patients and Rats with Epilepsy: A Pilot Study

**DOI:** 10.3390/biomedicines11123237

**Published:** 2023-12-07

**Authors:** Christopher Martínez-Aguirre, Luis Alfredo Márquez, Cindy Lizbeth Santiago-Castañeda, Francia Carmona-Cruz, Maria de los Angeles Nuñez-Lumbreras, Vladimir A. Martínez-Rojas, Mario Alonso-Vanegas, Gustavo Aguado-Carrillo, Norma L. Gómez-Víquez, Emilio J. Galván, Manola Cuéllar-Herrera, Luisa Rocha

**Affiliations:** 1Pharmacobiology Department, Center for Research and Advanced Studies, Mexico City 14330, Mexico; christopher_mtz@cinvestav.mx (C.M.-A.); luis.marquez@cinvestav.mx (L.A.M.); cindy.santiago@cinvestav.mx (C.L.S.-C.); fcarmona@cinvestav.mx (F.C.-C.); angeles_lumbreras@hotmail.com (M.d.l.A.N.-L.); vamartinezr@cinvestav.mx (V.A.M.-R.); nogomez@cinvestav.mx (N.L.G.-V.); egalvan@cinvestav.mx (E.J.G.); 2Center for Research on Aging, Center for Research and Advanced Studies, Mexico City 14330, Mexico; 3International Center for Epilepsy Surgery, HMG-Coyoacán Hospital, Mexico City 04380, Mexico; alonsomario@hotmail.com; 4Clinic of Epilepsy, General Hospital of México Dr. Eduardo Liceaga, Mexico City 06720, Mexico

**Keywords:** cannabidiol, glutamate, dual effects, drug-resistant epilepsy, temporal lobe epilepsy, extratemporal lobe epilepsy

## Abstract

Drug-resistant epilepsy (DRE) is associated with high extracellular levels of glutamate. Studies support the idea that cannabidiol (CBD) decreases glutamate over-release. This study focused on investigating whether CBD reduces the evoked glutamate release in cortical synaptic terminals obtained from patients with DRE as well as in a preclinical model of epilepsy. Synaptic terminals (synaptosomes) were obtained from the epileptic neocortex of patients with drug-resistant temporal lobe epilepsy (DR-TLE, *n* = 10) or drug-resistant extratemporal lobe epilepsy (DR-ETLE, *n* = 10) submitted to epilepsy surgery. Synaptosomes highly purified by Percoll-sucrose density gradient were characterized by confocal microscopy and Western blot. Synaptosomes were used to estimate the high KCl (33 mM)-evoked glutamate release in the presence of CBD at different concentrations. Our results revealed responsive tissue obtained from seven patients with DR-TLE and seven patients with DR-ETLE. Responsive tissue showed lower glutamate release (*p* < 0.05) when incubated with CBD at low concentrations (less than 100 µM) but not at higher concentrations. Tissue that was non-responsive to CBD (DR-TLE, *n* = 3 and DR-ELTE, *n* = 3) showed high glutamate release despite CBD exposure at different concentrations. Simultaneously, a block of the human epileptic neocortex was used to determine its viability through whole-cell and extracellular electrophysiological recordings. The electrophysiological evaluations supported that the responsive and non-responsive human epileptic neocortices used in the present study exhibited proper neuronal viability and stability to acquire electrophysiological responses. We also investigated whether the subchronic administration of CBD could reduce glutamate over-release in a preclinical model of temporal lobe epilepsy. Administration of CBD (200 mg/kg, p.o. every 24 h for 7 days) to rats with lithium-pilocarpine-evoked spontaneous recurrent seizures reduced glutamate over-release in the hippocampus. The present study revealed that acute exposure to low concentrations of CBD can reduce the glutamate over-release in synaptic terminals obtained from some patients with DRE. This effect is also evident when applied subchronically in rats with spontaneous recurrent seizures. An important finding was the identification of a group of patients that were non-responsive to CBD effects. Future studies are essential to identify biomarkers of responsiveness to CBD to control DRE.

## 1. Introduction

Epilepsy is a neurological disorder characterized by abnormal electrical activity in the brain and the expression of spontaneous recurrent seizures (SRS) [1]. Approximately one-third of patients with epilepsy present drug resistance to traditional antiseizure medications (ASMs) [2]. Drug-resistant epilepsy (DRE) is characterized by a failure to control seizures despite the use of the appropriate ASMs [3]. Preclinical and clinical evidence supports that DRE is related to an increase in the extracellular release of glutamate during the ictal and interictal periods [4,5,6]. This increase contributes significantly to the activation of neuroinflammatory processes, oxidative stress, and cell damage and death associated with excitotoxicity [7].

There is evidence that cannabidiol (CBD), a cannabinoid without psychoactive effects [8], induces anticonvulsant, anti-inflammatory, antioxidant, and anti-excitotoxic effects in the central nervous system (CNS) [9,10]. Acute exposure to CBD averted glutamate over-release from the synaptic terminals of animals submitted to cocaine-induced seizures [11]. Likewise, subchronic pretreatment with CBD reduced glutamate over-release and the sensorimotor disfunction induced short- and long-term after a severe traumatic brain injury in rats [12].

Clinical evidence supports the idea that chronic treatment with CBD combined with other ASMs reduces the frequency and severity of drug-resistant seizures associated with Dravet syndrome, Lennox–Gastaut syndrome, and tuberous sclerosis [13,14,15]. Regarding other types of epilepsies, one report showed that CBD treatment diminished the frequency of seizures and improved cognition in a patient with drug-resistant frontal lobe epilepsy [16]. Similarly, CBD in combination with ASMs reduced the frequency and severity of seizures in some patients with drug-resistant temporal lobe epilepsy (DR-TLE) [17]. Despite this evidence, it is important to note that some patients with DRE do not respond to the antiseizure effects of CBD [18]. The lack of response of these patients to CBD could be explained by an increased metabolism following oral administration that results in low bioavailability of this compound in the brain [19]. According to this rationale, we hypothesized that the direct exposure of brain tissue with epileptiform activity to CBD in patients with DRE results in inhibitory effects. The aim of this study was to investigate whether acute exposure to CBD reduces the glutamate over-release evoked by high potassium chloride (KCl) in synaptic terminals obtained from the human epileptic neocortex, a brain structure with a high glutamate content involved in seizure propagation [20]. We also investigated whether subchronic oral administration of CBD averts the hippocampal over-release of glutamate in vivo in a preclinical model of temporal lobe epilepsy.

## 2. Materials and Methods

### 2.1. Patients and Collection of Tissue

Neocortical epileptic tissue was obtained from patients with DR-TLE (group, *n* = 10) or extratemporal lobe epilepsy (DR-ETLE group, *n* = 10). The clinical data of the patients are summarized in Table 1. The patients underwent an extensive presurgical evaluation that included video electroencephalograms (EEGs) and magnetic resonance imaging (MRI). The scientific and ethics committees from the institutions involved in this study approved the protocol (055/2018). Written informed consent was obtained from all participants.

Neocortical tissue with epileptiform activity was collected during the epilepsy surgery (HUM-202 to HUM-219 from the International Center for Epilepsy Surgery, HMG-Coyoacán Hospital; HUM-220 to HUM-223 from the Clinic of Epilepsy, General Hospital of México “Dr. Eduardo Liceaga”). Immediately after resection, the tissue was immersed in an ice-cold isotonic buffer solution (sucrose 0.32 M, EDTA 1 mM, Tris-HCl 5 mM, pH 7.4) and was oxygenated by bubbling (0.5 L/h). Then, samples were transported from the surgery room to the laboratory, a transfer conducted in less than 45 min in order to maintain proper tissue viability. In the laboratory, the tissue was divided into two blocks and was used as follows: one block was used to estimate the glutamate release from synaptic terminals, while another block was evaluated for tissue viability through electrophysiological analysis. The experimental procedures were carried out simultaneously.

### 2.2. Evaluation of the Effect of CBD on the Evoked Glutamate Release in Synaptic Terminals Obtained from Patients with DRE

The evaluation of glutamate release can be performed in vivo through microdialysis experiments. However, due to the limitations of carrying out this type of technique in the human brain, we decided to evaluate the evoked glutamate release in isolated synaptic terminals (synaptosomes) in vitro. Synaptosomes have all the cellular machinery necessary to continue the production of ATP, contain cellular receptors, and maintain membranes potentially sensitive to depolarization [21].

#### 2.2.1. Isolation and Purification of Synaptosomes

A previously reported method was used to obtain the synaptosomes from brain tissue [21]. Briefly, human epileptic neocortex was placed in a buffer solution (sucrose 0.32 M, EDTA 1 mM, Tris-HCl 5 mM, pH 7.4) and was homogenized manually with a glass–Teflon homogenizer. The resulting suspension was centrifuged at 1000× *g* for 10 min at 4 °C. The supernatant was separated in a Percoll gradient (0, 3, 10, 15, and 23%). Then, tubes were centrifuged at 31,000× *g* for 5 min at 4 °C. The purified synaptosomal fraction, located between phase 10% and 15% of the gradient, was diluted in isotonic sucrose solution and centrifuged at 20,000× *g* for 30 min at 4 °C to remove Percoll remnants. The obtained pellet was resuspended in artificial cerebrospinal fluid (ACSF, containing, in mM, 125 NaCl, 2.5 KCl, 5 Na_2_HPO_4_, 0.5 NaHCO_3_, 1 MgCl_2_, 0.2 ascorbic acid, and 1.2 CaCl_2_) and divided into three independent fractions used for the following purposes: (a) characterization of synaptosomes by confocal microscopy; (b) evaluation of the expression of synaptophysin and postsynaptic density protein-95 (PSD-95) by Western blot; and (c) analysis of the glutamate release evoked by high KCl.

#### 2.2.2. Characterization of Synaptosomes by Confocal Microscopy

Synaptosome preparation was incubated in 10 μM of di-8 amino-naphthyl-ethenyl-pyridinium dye (di-8-ANEPPS Invitrogen, Cat. D3167, ThermoFisher, Waltham, MA, USA) at 24 °C for 5 min in order to detect changes in membrane voltage. Then, the excess dye was washed out with ACSF, and the preparation was centrifuged at 500× *g* for 5 min. Thereafter, the pellet was resuspended with ACSF. Samples were placed in a perfusion chamber and were analyzed using a confocal microscope (Zeiss Airyscan, LSM800; 62× objective).

#### 2.2.3. Analysis of Protein Expression of Synaptophysin and Postsynaptic Density Protein 95 in Human Synaptosomes

Synaptophysin and postsynaptic density protein 95 (PSD-95) are two proteins expressed at synapses that participate in the exocytosis of neurotransmitters and in the establishment of synapsis, respectively. Hence, the expression of these two proteins was evaluated in order to validate synaptosome preparations [22].

The synaptosome preparation was centrifuged at 21,000× *g*, and the pellet was homogenized in radioimmunoprecipitation lysis buffer (RIPA buffer, 50 mM Tris–HCl, 150 mM NaCl, 1 mM EDTA, and 0.1% Triton X-100, pH 7.5) and a mixture of protease inhibitors (Complete Roche Diagnostics GmbH, Mannheim, Germany). Samples (30 μg/lane) were loaded in Laemmle sample buffer (cat. 1610737, Bio-Rad Laboratories, Hercules, CA, USA) and were separated in 12% sodium dodecyl sulfate (SDS)–polyacrylamide gel electrophoresis gel (85 V for 30 min and 100 V for 2 h) using a running buffer (25 mM Tris, 192 mM glycine and 0.1% SDS, pH 8.3; cat. 1610723, Bio-Rad Laboratories, Hercules, CA, USA). Subsequently, the proteins were transferred onto polyvinylidene difluoride membranes (Immun-blot, cat. 1620264, Bio-Rad Laboratories, Hercules, CA, USA). After electroblotting, membranes were incubated for 1 h at room temperature with 5% blocking solution (Blot-QuickBlocker, cat. WB57, EMD Millipore, Burlington, MA, USA) dissolved in TBS-T buffer (20 mM Tris, 500 mM NaCl, 0.1% Tween 20, pH 7.5).

Subsequently, membranes were incubated overnight at 4 °C with the corresponding primary antibody (Synaptophysin, Abcam, Waltham, MA, USA, Cat. AB8049; PSD-95, Invitrogen, Rockford, IL, USA, Cat. 7E3-1B8, ThermoFisher). Membranes were washed with TBS-T and incubated for 2 h with horseradish peroxidase–coupled secondary antibody. Protein signal detection was achieved with Clarity Western ECL Blotting Substrates (Bio-Rad Laboratories, Hercules, CA, USA) according to the manufacturer’s indications. Scanning of the immunoblots was performed, and the bands were quantified by densitometry using an image analysis program (Image Lab Software Version 5.2.1, Bio-Rad, Hercules, CA, USA). Each sample was evaluated in triplicate.

#### 2.2.4. Evaluation of Glutamate Release Evoked by High KCl in Human Synaptosomes

The synaptosome sample was subdivided into seven fractions in order to simultaneously determine the glutamate release under different experimental conditions, as follows:(a)Evoked glutamate release in the presence of CBD. Independent aliquots were incubated with CBD at different concentrations (100 nM, 1 µM, 10 µM, 100 µM, and 1 mM) at 24 °C for 15 min. Then, KCl was added to the preparation to achieve a final concentration of 33 mM and was incubated at 24 °C for 2 min. Afterward, the samples were centrifuged at 21,000× *g* for 5 min at 4 °C. The supernatant was collected and stored at −80 °C until the analysis of glutamate release by HPLC (see below).(b)Evoked glutamate release in the absence of CBD. A parallel fraction from the same tissue was handled as indicated previously, except that the incubation was carried out with 50 µL of sodium dodecyl sulfate (DMSO 0.5%, vehicle of CBD) instead of CBD.(c)Basal glutamate release. An independent aliquot was manipulated similarly to the fraction used in (b), except that it was incubated with 5 µL of ASCF instead of KCl 33 mM.

#### 2.2.5. Quantification of Glutamate by High-Performance Liquid Chromatography (HPLC)

Five microliters obtained from the supernatant fraction of the samples previously submitted to the different experimental conditions (see Section 2.2.4) were added to 10 µL of OPA-NAC solution (*O*-Phthalaldehyde 0.01 M, *N*-acetylcysteine 0.01 M, in methanol-sodium tetraborate 0.1 M at a range 1:5, final pH 9.3 ± 0.05). This preparation was injected into the chromatograph using a fluorescence detector (excitation λ of 250 nm and emission λ of 395 nm) model Waters^®^ 474 and a 3.9 × 150 mm column (Nova-Pack, 4 µm, C18, Waters, Milford, MA, USA) reverse-phase chromatography (separation and quantification). The mobile phase consisted of solution A (concentrate of acetates buffer and water HPLC, 1:10; the concentrate was 1.4 M sodium acetate, 10 mM EDTA disodium salt, 77 mM triethylamine, adjusted to pH 5.05 with phosphoric acid), solution B (acetonitrile HPLC), and solution C (water HPLC), which was used at a gradient flow rate 1.0 mL/min. The glutamate concentration was expressed in molarity units per mg of protein estimated using the Bradford protocol in parallel fractions [23].

### 2.3. Evaluation of Viability of the Neocortical Epileptic Tissue by In Vitro Electrophysiology

This experiment was designed to assess the viability of neocortical epileptic tissue obtained from patients with DRE. The experimental procedure focused on corroborating that the results obtained from the synaptosomes were due to the intrinsic properties of the sample and not to the loss of viability of the evaluated human brain tissue.

A block of neocortical tissue (≈1.5 × 2.6 cm, length and width, respectively, and ≈0.75 cm deep) was delimited with the help of a scalpel. The tissue block was glued onto the platform of a vibratome tissue slicer (Leica VT1000S; Nussloch, Germany) and was then sliced at 400 µm thickness. The brain slices were stabilized at 34 °C for 30 min in an ACSF solution containing (in mM) 125 NaCl, 2.5 KCl, 1.25 Na_2_HPO_4_, 25 NaHCO_3_, 4 MgCl_2_, 1 CaCl_2_, and 10 D-glucose. Then, the slices were maintained at room temperature for at least 90 min before any experimental procedure. An individual slice was transferred to a submerged chamber (total volume: 400 µL) and was perfused with ACSF containing (in mM) 125 NaCl, 2.5 KCl, 1.25 Na_2_HPO_4_, 25 NaHCO_3_, 2 MgCl_2_, 2.5 CaCl_2_, and 10 D-glucose, with pH ≈ 7.30–7.35. Whole cell patch-clamp and extracellular recordings were performed simultaneously at 33 ± 1 °C in parallel slices of the neocortical tissue.

#### 2.3.1. Whole Cell Recordings

The neocortical slices were visualized using infrared differential interference contrast optics coupled to an FN1 Eclipse microscope (Nikon Corporation, Minato, Tokyo, Japan). Pyramidal cells, located in layer V, were identified based on shape and position. The patch pipettes were pulled from borosilicate glass using a micropipette puller (P97, Sutter Instruments, Novato, CA, USA). The pipette tips had a resistance of 5–8 MΩ when filled with an intracellular solution with the following composition (in mM): 135 K^+^-gluconate, 10 KCl, 5 NaCl, 1 ethylene glycol-bis(β-aminoethyl ether)-*N*,*N*,*N*′,*N*′-tetraacetic acid (EGTA), 10 *N*-(2-hydroxyethyl)piperazine-*N*′-(2-ethanesulfonic acid) (HEPES), 2 Mg^2+^-ATP, 0.4 Na^+^-GTP, and 10 phosphocreatine, with pH ≈ 7.20–7.28. The patch-clamp recordings were performed with an Axopatch 200B amplifier (Molecular Devices, San José, CA, USA), digitized at 40 kHz, and filtered at 1 kHz with a Digidata 1550B (Axon Instruments, Palo Alto, CA, USA). Digital signals were acquired and analyzed offline with the help of pCLAMP 11.2 software (Molecular Devices).

#### 2.3.2. Electrophysiological Evaluation

The resting membrane potential (RMP) was measured after the initial break-in from giga-seal to the whole-cell configuration in voltage-clamp mode. Once determined, the amplifier’s configuration was switched to current-clamp mode. A series of current injections (1 s duration, 30 pA increments) was applied to determine the current–voltage relationship (I–V relationship), somatic input resistance (RN), membrane time constant (τmemb), and membrane capacitance. The somatic RN was calculated as the slope value of the first-degree polynomial function:(1)$$f\left(x\right)=mx+b$$
fitted to the I–V relationship around the RMP. The τmemb was calculated by fitting a single exponential function
(2)$$\left(f\left(t\right)=\sum_{i=1}^{n}A_{i}\;\;e^{-\frac{t}{\tau i}}+C\right)$$
to a voltage response (−30 pA) that favored passive membrane charge/discharge.

#### 2.3.3. Extracellular Recordings

The field excitatory postsynaptic potentials (fEPSP) were recorded in layer V using borosilicate pipettes filled with 3 M NaCl solution. When filled with this solution, the tip resistance of the pipette was 1–2 MΩ. The test stimuli (0.067 Hz, 100 µs) were delivered via a custom-made nichrome electrode (38 µM bare diameter) placed in layer I of the neocortical slice. The stimulation electrode and recording pipette followed the columnar arrangement of the cortex to elicit orthodromic fEPSPs. The electrophysiological responses were amplified with a Dagan BVC-700 A amplifier (Minneapolis, MN, USA) coupled with an extracellular headstage (Dagan, model 8024) and high-pass filtered at 0.3 Hz. The current pulses used for synaptic responses were delivered via a high-voltage isolation unit (A365D; World Precision Instruments, Sarasota, FL, USA) controlled with a Master-8 pulse generator (AMPI, Jerusalem, Israel). A Hum Bug noise eliminator (Quest Scientific Instruments, North Vancouver, BC, Canada) was also used to achieve additional electrical noise cancelation. The paired-pulse ratio was expressed as the ratio between the pair’s second and first fEPSP (S2/S1).

### 2.4. Evaluation of the Effect of Subchronic Treatment with CBD on Glutamate Over-Release in the Hippocampus of Rats with Temporal Lobe Epilepsy

Studies support that clinical effects of CBD in epilepsy are evident after its repetitive administration [24]. This experiment was designed to determine whether subchronic treatment with CBD modified the interictal over-release of glutamate in the hippocampus of rats with SRS. CBD was administrated at 200 mg/kg p.o. since this dose proved to reduce the severity and the consequences of seizures in different preclinical models [10].

#### 2.4.1. Animals

Male Wistar rats (250–300 g) were used. The animals were maintained in controlled conditions of temperature (20–22 °C) and humidity (50–60%) with 12-light/dark cycles, with food and water ad libitum. The experimental protocol was conducted according to the Official Mexican Standard (NOM-062-ZOO-1999) and was approved by the Ethics Committee of the Center for Research and Advanced Studies (CICUAL, 125-15).

#### 2.4.2. Induction of Status Epilepticus and Spontaneous Recurrent Seizures by Lithium-Pilocarpine

Rats were habituated to handling by daily administration of saline solution (1 mL/kg, i.p.) for five consecutive days. Twenty-four hours after the last day of habituation, the rats received lithium chloride (3 mEq/kg, i.p.). Eighteen hours later, methyl scopolamine (1 mg/kg, s.c.), a muscarinic antagonist, was administered to minimize the peripheral effects of pilocarpine. After 30 min, pilocarpine (30 mg/kg, i.p.) was injected with the purpose of inducing status epilepticus (SE). The onset of SE was determined when animals showed continuous motor seizures for more than 2 min without recovery. Diazepam was applied at 2 and 10 h after the onset of SE (2.5 and 1.25 mg/kg, i.m., respectively) to reduce the convulsive activity and decrease the mortality rate. After the induction of SE, the animals were kept in recovery and were video recorded to determine the onset of SRS.

One week after experiencing the first SRS, the rats were anesthetized with a mixture of Zoletil-50^®^ and xylazine (10 mg/kg, i.m., and 12 mg/kg, i.m., respectively). Subsequently, they were placed in a stereotactic frame, and a guide cannula was implanted at the level of the cortex and on the left hippocampus according to the following coordinates: −5.3 mm anteroposterior with reference to Bregma; 5.2 mm lateral; 3 mm depth with reference to the cranial surface. Four stainless steel screws were implanted in the skull over the frontal, parietal, and cerebellar cortex to support the implant. All elements were fixed to the skull with dental acrylic. Rats were included in one experimental group one week after the surgery.

#### 2.4.3. Experimental Groups

(a)SRS-CBD group (*n* = 7). Rats received CBD 200 mg/kg, p.o. every 24 h for 7 days. Twenty-four hours after the last administration of CBD, the animals were subjected to microdialysis experiments to determine the interictal extracellular levels of glutamate in the hippocampus. A microdialysis cannula designed according to Maidment et al., 1989 [25], with a 3 mm active part of polyacrylonitrile (40 kDa pore), was inserted through the guide cannula and implanted in the left dorso-ventral hippocampus. The microdialysis cannula was constantly perfused with fresh and sterile ACSF (see Section 2.2.1) at a flow rate of 2 µL/min. Two hours after the probe implantation, recovery of the dialysates was carried out every 30 min for 2 h. Dialysates were processed for glutamate quantification by HPLC as previously described (see Section 2.2.5). Twenty-four hours after the end of the microdialysis experiment, rats were sacrificed with an overdose of pentobarbital (70 mg/kg i.p.), and their brains were used to determine the location of the microdialysis probes by Nissl staining.(b)SRS-Vh group (*n* = 7). The animals were handled in a similar way to the SRS-CBD group, with the exception that they received a vehicle (coconut oil 9.52 mL/kg, p.o.) instead of CBD.(c)Sham-CBD group (*n* = 7). Rats were handled similarly to the SRS-CBD group, except that saline (1 mL/kg, i.p.) was applied instead of lithium chloride, methyl scopolamine, and pilocarpine.(d)Sham-Vh group (*n* = 7). The animals were handled in a similar way to the Sham-CBD group, with the exception that a vehicle (coconut oil 9.52 mL/kg, p.o.) was administered instead of CBD.

### 2.5. Statistical Analysis

A Shapiro–Wilk test was performed to assess the normality of the dataset. The effect of CBD on glutamate release was evaluated using one-way ANOVA, followed by Tukey’s test. The electrophysiological responses were analyzed using Student’s *t*-test.

## 3. Results

### 3.1. Characterization of Synaptosomes

Confocal microscopy images revealed small clusters of spherical elements stained with di-8-ANEPPS. These spheres were less than 1 µm in diameter (Figure 1A). The Western blot experiments showed the expression of synaptophysin and PSD-95 proteins in the samples obtained from centrifugations (Figure 1B). This group of evidence supports that samples isolated from the neocortical epileptic tissue of patients with DRE corresponded to synaptosomes.

### 3.2. Effects of CBD on Glutamate Release from Synaptic Terminals from Patients with DRE

This experiment focused on determining whether acute exposure to CBD reduces the glutamate-evoked release in synaptosomes obtained from patients with DRE. The results obtained revealed brain tissue responsive and non-responsive to CBD.

#### 3.2.1. Effects of CBD on Evoked Glutamate Release in Responsive Human Epileptic Neocortex of Patients with DRE

The results obtained from the DR-TLE group revealed seven samples (70%) responsive to CBD; i.e., CBD was able to modify the glutamate-evoked release. The basal extrasynaptosomal concentration of these samples was 39.2 ± 4.4 nmol/mg of protein. The incubation with high KCl increased the extrasynaptosomal glutamate concentration to 137.6 ± 20.4 nmol/mg of protein (*p* = 0.0004 vs. basal). The extrasynaptosomal glutamate concentration evoked by high-KCl was lower when synaptosomes were pre-incubated with CBD 100 nM, 1 µM, 10 µM, or 100 µM (50.7%, *p* = 0.0153; 52.3%, *p* = 0.0103; 54%, *p* = 0.0063; 48.4%, *p* = 0.0167, lower respectively) vs. high-KCl alone. This effect was not evident when synaptosomes were preincubated at a higher concentration of CBD (1 mM, 15.5%, *p* = 0.8125 lower vs. high-KCl alone) (Figure 2A).

Concerning the DR-ETLE group, the synaptosomes obtained from seven patients (70%) were responsive when exposed to CBD. This group showed a basal extrasynaptosomal glutamate concentration of 52.5 ± 5.8 nmol/mg of protein (*p* = 0.1023 vs. responsive DR-TLE). The exposure to high-KCl augmented the extrasynaptosomal glutamate concentration to 172.8 ± 21.4 nmol/mg of protein (*p* = 0.0022 vs. basal; *p* = 0.2639 vs. responsive DR-TLE). In contrast to high-KCl alone, synaptosomes from the responsive DR-ETLE group pre-incubated with CBD showed lower evoked glutamate release (100 nM, 64.0%, *p* = 0.0054; 1 µM, 61.8%, *p* = 0.0261; 10 µM, 58.1%, *p* = 0.0132, lower vs. high-KCl alone). However, preincubation with CBD at 100 µM or 1 mM did not change the evoked extrasynaptosomal glutamate concentration (15.3%, *p* = 0.9050; 12.9%, *p* = 0.9612, lower respectively, vs. high-KCl alone) (Figure 2B).

Patients responsive to CBD showed the following clinical characteristics: (a) patients with DR-TLE (HUM-202, HUM-203, HUM-205, HUM-209, HUM-211, HUM-220, and HUM-221) were 25.1 ± 1.9 years old; 11.3 ± 3.3 years of seizure onset age; 13.9 ± 3.6 years of epilepsy duration; 22.6 ± 11.7 seizures per month. Patients with DR-ETLE (HUM-206, HUM-207, HUM-210, HUM-213, HUM-217, HUM-218, and HUM-219) were 16.0 ± 3.5 years old (*p* = 0.0416 vs. responsive DR-TLE); 5.8 ± 2.3 years of seizure onset age (*p* = 0.1970 vs. responsive DR-TLE); 10.2 ± 2.1 years of epilepsy duration (*p* = 0.3984 vs. responsive DR-TLE); 6.6 ± 3.4 seizures per month (*p* = 0.2920 vs. responsive DR-TLE) (Table 1).

#### 3.2.2. CBD Fails to Modify the Evoked Glutamate Release in Non-Responsive Human Epileptic Neocortex

The results obtained revealed that CBD exposure did not induce effects on the glutamate-evoked release of three (30%) neocortices of the DR-TLE group and three (30%) neocortices of the DR-ETLE group. All of these patients were included in the non-responsive (N-R) group since no significant differences were found between groups.

The extrasynaptosomal concentration of glutamate under basal conditions in the N-R group was 37.3 ± 5.4 nmol/mg of protein (*p* = 0.7924 vs. responsive DR-TLE; *p* = 0.0846 vs. responsive DR-ETLE). High-KCl increased the extrasynaptosomal glutamate concentration to 93.4 ± 10.5 nmol/mg of protein (*p* = 0.0105 vs. basal). This value was significantly lower when compared to responsive DR-TLE (50.7% lower, *p* = 0.0829) and responsive DR-ETLE (46.0% lower, *p* = 0.0093) groups under the same experimental conditions. The extrasynaptosomal glutamate release evoked by high-KCl in fractions preincubated with different concentrations of CBD remained similar to those incubated with high-KCl alone (2.8% lower, *p* > 0.9999; 18.3% lower, *p* > 0.9999; 26% lower, *p* > 0.9999, 8.8% lower, *p* > 0.9999, 22.9% higher, *p* > 0.9999, for CBD 100 nM, 1 µM, 10 µM, 100 µM, and 1 mM, respectively, vs. high-KCl alone) (Figure 2C).

Patients from the N-R group (HUM-204, HUM-222, and HUM-223 from DR-TLE, and HUM-208, HUM-212, and HUM-214 from DR-ETLE group) presented the following clinical characteristics: 14.8 ± 5.0 years of age; 8.7 ± 5.3 years of seizure onset age; 6.0 ± 2.4 years of epilepsy duration; 30.8 ± 23.9 seizures per month (Table 1). No significant differences were detected in the clinical variables when comparing N-R patients from the DR-TLE and DR-ETLE groups. In addition, no differences were found when comparing responsive vs. N-R groups (*p* > 0.05).

### 3.3. Neocortical Tissue of Patients with DRE Keeps Its Viability and Electrophysiological Properties after Surgery

We analyzed the electrophysiological properties of neocortical neurons obtained from patients from the responsive DR-ETLE group and from the N-R group in order to corroborate its viability after surgery.

#### 3.3.1. Whole Cell Recording

Whole cell patch-clamp and extracellular recordings were performed in acute slices to explore possible differences in the passive and active electrophysiological properties as well as possible changes in the synaptic strength of the human tissue from the responsive DR-ETLE vs. the N-R group.

Somatic whole-cell patch-clamp recordings were performed in neurons obtained from five different slices located in layer (L) V of the prefrontal cortex. After the initial somatic break-in in voltage-clamp mode, the neurons were switched to current-clamp, and passive and active electrophysiological properties were measured. Figure 3A1,A2 show representative current-voltage relationships (1-s duration/30 pA increments) obtained from both groups of neurons. The resting membrane potential in the responsive DR-ETLE group was −68.3 ± 1.2 mV, whereas in the N-R group, it was −78.4 ± 0.7 mV (*p* < 0.001 vs. responsive DR-ETLE; Figure 3B1). Also, the depolarizing current steps (60–90 pA) revealed a burst of action potentials (two to three events) and a depolarizing membrane plateau that was slightly larger in the N-R compared to the responsive DR-ETLE group (see arrowheads in Figure 3A1,A2). Next, we determined the somatic input resistance of both groups. For the responsive DR-ETLE group, the RN was 137.6 ± 9.5 MΩ, whereas for the N-R group, the RN was 163.3 ± 29.3 MΩ (*p* = 0.4283 vs. responsive DR-ETLE; Figure 3B2). In addition, with the injection of the low-magnitude negative current steps, we determined the membrane time constant (τmemb). While the responsive DR-ETLE had a τmemb of 24.5 ± 1.8 ms, the τmemb of the N-R was 20.4 ± 2.3 ms (*p* = 0.1980 vs. responsive DR-ETLE; Figure 3B3). Last, we determined the membrane capacitance of both experimental groups. For the responsive DR-ETLE group, the capacitance was 178.1 ± 3.4 pF, while the capacitance of the N-R group was 142 ± 32.1 pF (*p* = 0.2959 vs. responsive DR-ETLE; Figure 3B4). The whole cell patch-clamp recordings indicate no differences in the passive and active electrophysiological properties of human cortical tissue responsive and non-responsive to CBD.

#### 3.3.2. Extracellular Recordings

Next, in another group of slices, we performed extracellular recordings to maximize the measurement of the synaptic activity from both experimental groups. A stimulation electrode was placed on layer I/II, and the resulting field excitatory postsynaptic potential (fEPSP) was recorded in layer V. We observed that the magnitude of the fEPSP slope was stable for up to 20 min in response to constant stimulation (0.067 Hz, 100 µs), as illustrated in Figure 3C (fEPSP, *n* = 4 slices). The synaptic stability was found to be consistent in both experimental conditions. The stability of the recorded fEPSP is depicted in the cumulative probability graph in Figure 3D. Give that the recording of fEPSPs was performed with paired pulses of electrical stimulation (60 ms interstimulus interval), we determined the paired-pulse ratio of the synaptic responses of the human cortex. Consistent with typical paired-pulse facilitation (PPF) from the cortical LI/II-LV synapse [26], the cortical slices exhibited PPF (Figure 3E), corroborating a low probability of presynaptic release from cortical LI/LV synapse.

The extracellular recordings supported that the human neocortex used in the present study exhibits neuronal viability and the stability to acquire electrophysiological responses. These results were independent of the ability to respond to the pharmacological effects of CBD on the evoked glutamate release.

### 3.4. Effect of Subchronic Administration with CBD on Interictal Glutamate Release in the Hippocampus of Rats with Epilepsy

The results obtained from human synaptosomes indicate that acute exposure to CBD may modify the evoked release of glutamate. On the other hand, clinical evidence supports that CBD effectiveness is achieved after its repetitive administration [14]. An experiment using a preclinical model of TLE was designed to determine whether subchronic administration of CBD reduces glutamate over-release in the epileptic hippocampus. We used rats with SRS previously submitted to pilocarpine-induced SE. This preclinical model shares the characteristics of TLE, and about 40% of the rats develop drug-resistant seizures [27,28].

The extracellular concentration of glutamate in the hippocampus of rats from the Sham-Vh group was 1.22 ± 0.31 µM. The Sham-CBD group presented a hippocampal concentration of glutamate similar to the Sham-Vh group (1.08 ± 0.31 µM, *p* = 0.9994). In the SRS-Vh group, the interictal extracellular release of glutamate was higher when compared to the Sham-Vh group (319%, *p* = 0.0107 vs. Sham-Vh). The interictal extracellular release of glutamate obtained from the SRS-CBD group was similar to the Sham-Vh group (*p* = 0.9892) and lower (72%, *p* = 0.0168) when compared to the SRS-Vh group (*p* = 0.0168) (Figure 4).

The histological evaluation confirmed that the microdialysis probe was implanted in the left hippocampus of all the animals used for this experiment.

## 4. Discussion

This study focused on evaluating CBD effects in glutamate over-release associated with epilepsy. Initially, we analyzed the effects of acute exposure to CBD on the evoked glutamate release in synaptic terminals of human epileptic neocortex obtained from epilepsy surgery. Our results revealed responsive and non-responsive tissue to CBD. Responsiveness to CBD was associated with neither the type of epilepsy nor the clinical factors of the patients. In addition, we found that subchronic administration with CBD averted glutamate over-release in the hippocampus of rats with epilepsy.

Studies support that the extracellular levels of glutamate correlate with the tissue content of this amino acid in epileptic brain areas of patients with DRE, both in vivo [20,29] and in vitro, using synaptosomes [30]. The results obtained from the present study indicated that neocortices responsive and non-responsive to CBD presented similar glutamate release under basal conditions. In addition, electrophysiological evaluation supported that all human epileptic neocortices evaluated in the present study were viable throughout the experimental period; thus, responsiveness to CBD did not depend on the tissue viability.

CBD has been shown to be effective in reducing the frequency and severity of seizures in different types of epilepsies [13,31]. Preclinical evidence indicates that CBD exerts its antiseizure effects by decreasing neuronal excitability [32]. Our results revealed that acute exposure to low concentrations of CBD reduces evoked glutamate release at the synaptic terminals of some patients with DR-TLE or DR-ETLE. This effect can be explained by the action of CBD on different receptors with inhibitory effects [33]. CBD shifts the voltage dependence of Kv7.2/7.3 channels in the hyperpolarizing direction, reducing neuronal excitability [34]. CBD also blocks currents mediated by Ca(V)3.1, Ca(V)3.2, and Ca(V)3.3 channels expressed in human embryonic kidney 293 cells [35]. Since glutamate release is a process highly mediated by calcium, CBD could decrease the release of glutamate to the extrasynaptosomal space by reducing the calcium currents [36].

An interesting finding was that the inhibitory effect mediated by CBD at low concentrations in responsive brain tissue was not evident at high concentrations. Dual effects of CBD have been detected in other studies. Administration of CBD results in dual effects in a mouse model of schizophrenia [37]. In a simulated public speaking test, the administration of 300 mg of CBD reduced the visual analog mood scale (a measure of anxiety), an effect less evident with 600 mg [38]. The absence of inhibitory effects at high concentrations of CBD could suggest an excitatory effect mediated by this fitodrug. Concerning this issue, it is known that CBD may block CB1 receptors [39] with a consequent increase in the intrasynaptosomal concentration of calcium and facilitation of glutamate release [40].

Previous clinical studies support that some patients with DRE do not respond to treatment with CBD, i.e., the treatment does not reduce the frequency and/or severity of seizures [17,18]. Our experiments allowed for the identification of epileptic neocortices non-responsive to CBD from patients with DRE. The lack of responsiveness of some tissue samples in the present study can be explained by the loss of viability [41,42]. However, the electrophysiological evaluation supported the viability of responsive and non-responsive tissue samples. In contrast to responsive tissue, non-responsive tissue showed low glutamate release evoked by high-KCl. This result correlated with low resting potential detected throughout the electrophysiological evaluation, suggesting lower neuronal excitability in the non-responsive tissue [43]. These results suggest that the effect of CBD depends on the rate of neuronal excitability. Future studies are essential to support this notion.

The lack of effect of CBD in some epileptic neocortices can be explained by changes in targets. Indeed, studies support that some targets of CBD, such as α-amino-3-hydroxy-5-methyl-4-isoxazolepropionic acid (AMPA), *N*-methyl-D-aspartic acid (NMDA), kainate, muscarinic M_2_, 5-HT_1A_ and GABA_B_ receptors are modified by DRE [44,45]. The lower responsiveness of some patients to CBD could also be associated with changes in the subcellular localization of receptors through which CBD exerts its effects, such as GABA_A_, NMDA, and AMPA, among others [33,46]. Lipid rafts, cell membrane regions enriched in specific lipids, are involved in synaptic transmission, propagation of action potential, and cell signaling [47]. Translocation of GABA_A_ receptors into lipid rafts reduces the signal transduction mediated by their activation, affecting the effect of ASMs acting on GABAergic neurotransmission, such as vigabatrin and benzodiazepines [48,49]. Moreover, there is evidence that activation of NMDA receptors in cultured hippocampal neurons recruits AMPA receptors to raft domains in the cell surface, increasing the probability of activation [50].

Changes in the structure of receptors and/or ion channels as a consequence of genetic factors represent an explanation for the lack of effect of CBD in the brain tissue of some patients with DRE. This hypothesis highlights the relevance of pharmacogenetics studies in patients with DRE, especially in those non-responsive to CBD [51,52,53].

In our study, subchronic treatment with CBD averted glutamate over-release in rats with SRS. It has been described that chronic treatments with CBD in patients produce antiseizure effects reducing the frequency and severity of the ictal events [54]. Santiago–Castañeda et al. reported that subchronic pretreatment with CBD decreased the glutamate over-release consequence of a severe traumatic brain injury [12]. In addition, chronic treatment with CBD (60 mg/kg p.o.) in rats submitted to a pentylenetetrazole-kindling model reduced mortality and cognitive dysfunction and increased the latency to the first generalized seizure [55]. This effect could be related to the activation of adenosine and CB_2_ receptors, as reported in an in vitro model of newborn hypoxic-ischemic brain damage in which acute exposure to CBD reduced glutamate release [56]. Similarly, it has been suggested that CBD can limit neuronal excitability by reducing presynaptic intracellular calcium concentrations mediated by transient receptor potential vanilloid 1 (TRPV1) and GPR55, therefore preventing glutamate over-release [57,58]. In addition, CBD could also be modifying the expression and signaling pathways of NMDA receptors and CREB/BDNF [59]. It is possible to suggest that these changes are more evident after the subchronic administration of CBD. However, more studies are needed since the information about the effects of chronic administration of CBD is currently limited. Similarly, it is important to assess whether the protective effects of CBD against evoked glutamate over-release reduce the brain damage and cell death that occurs in epilepsy [60].

One limitation of the present study is the small sample size of human brain tissue, which makes it difficult to detect the influence of clinical factors as well as ASMs on the response to CBD.

## 5. Conclusions

Our studies support that acute exposure to low concentrations of CBD reduces evoked glutamate release from synaptic terminals obtained from some patients with DRE, an effect not evident at high concentrations. The lack of effect of non-responsive tissue indicates that some patients are not able to respond to the CBD effects. Electrophysiological experiments showed that the neocortical tissue was viable, suggesting that the lack of response to CBD is due to the inherent properties of the samples. The subchronic administration of CBD averted in vivo glutamate over-release in a preclinical model of epilepsy, suggesting a good outcome following the administration schemes used in the clinic. Taken together, our results highlight the importance of identifying biomarkers associated with the responsiveness to CBD treatment, as well as rigorous control of the doses of CBD used in the treatment of DRE.

## Figures and Tables

**Figure 1 biomedicines-11-03237-f001:**
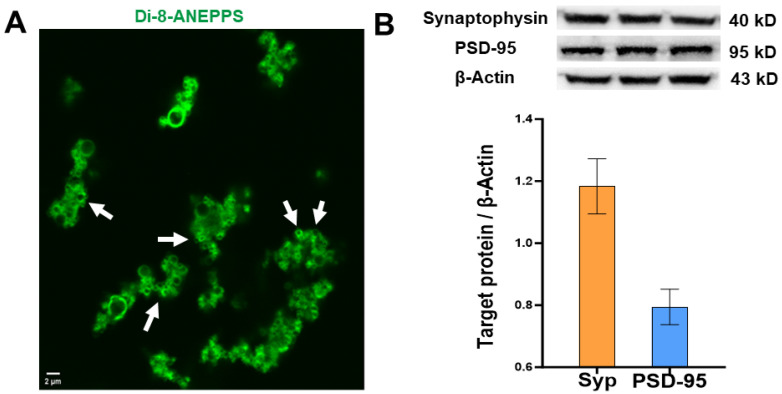
Characterization of synaptosomes obtained from the epileptic neocortex of patients with drug-resistant epilepsy. (**A**) Representative image of membrane stained with Di-8-ANEPPS. White arrows indicate synaptosome membranes. (**B**) Synaptophysin and PSD-95 protein expression relative to β-Actin in brain samples obtained from patients with drug-resistant epilepsy. The upper panels show representative Western blots of Synaptophysin, PSD-95, and β-Actin proteins of brain samples obtained from three patients.

**Figure 2 biomedicines-11-03237-f002:**
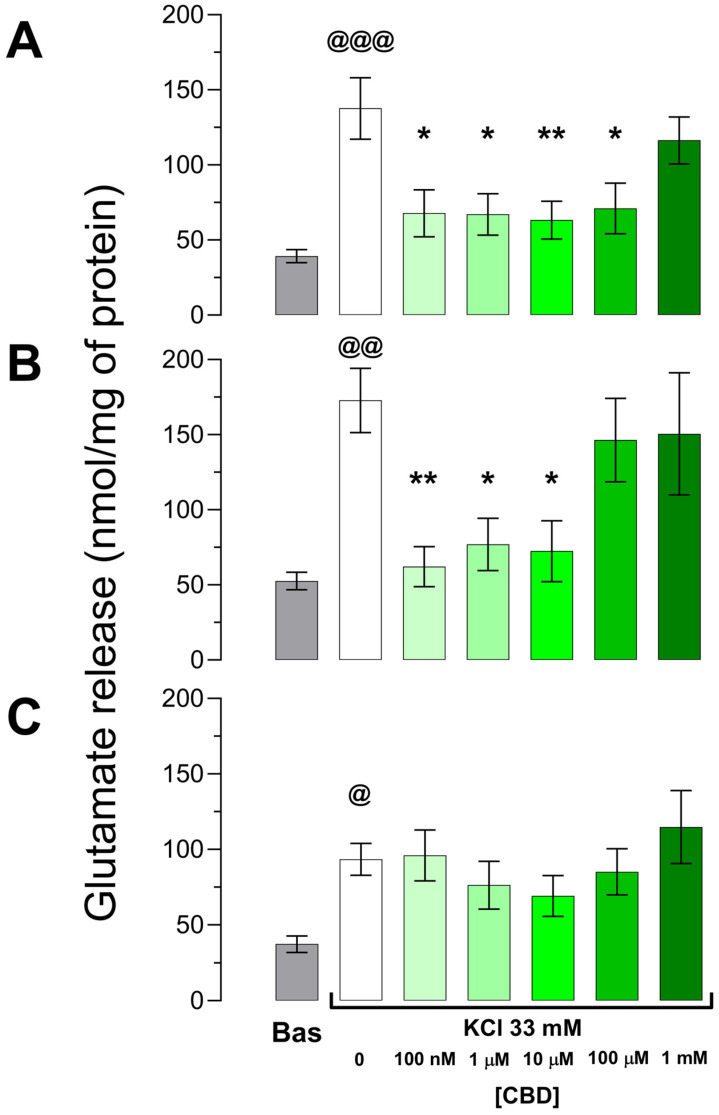
Glutamate-evoked release from synaptosomes obtained from responsive epileptic neocortex to cannabidiol (CBD) of (**A**) patients with drug-resistant temporal lobe epilepsy, (**B**) patients with drug-resistant extratemporal lobe epilepsy, and (**C**) from non-responsive epileptic neocortices to CBD. Note that under basal conditions, glutamate release is similar in all three groups. High potassium depolarization evoked an increase in the extrasynaptosomal glutamate concentration. The increase was avoided when synaptosomes were preincubated with low doses of CBD both in temporal (100 nM to 100 µM, *n* = 7) and extratemporal DRE (100 nM to 10 µM, *n* = 7). However, this effect was not evident in the tissue obtained from six patients (panel C, *n* = 6). Bas: basal. One-way ANOVA followed by Tukey’s test. @ *p* < 0.05, @@ *p* < 0.01, @@@ *p* < 0.001 vs. basal; * *p* < 0.05, ** *p* < 0.01 vs. high-KCl alone.

**Figure 3 biomedicines-11-03237-f003:**
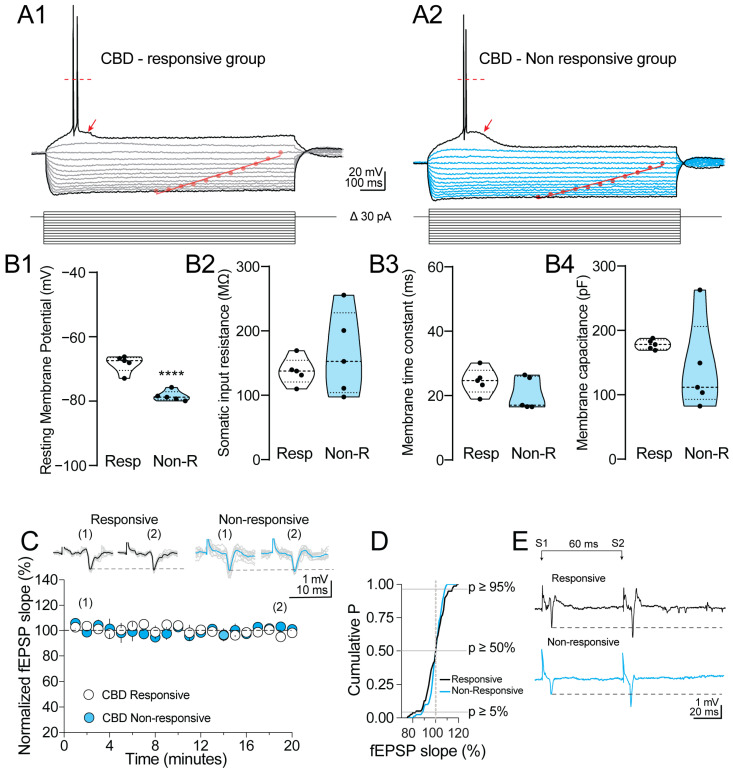
Electrophysiological properties of human cortical neurons. (**A1**,**A2**) Representative current–voltage relationships obtained from the CBD-responsive and CBD-non-responsive group. The I-V curves were performed at the resting membrane potential of each cell. CBD-responsive DR-ETLE was slightly depolarized. Notice that the injection of depolarization pulses revealed a short bursting of action potentials and a membrane plateau (see arrowheads). The dashed line over the spikes represents the action potential overshoot. The scatter lines within the voltage deflections represent the I-V linear adjustment. (**B1**) Violin graphs showing the average RMP for each group (5 neurons per group). The symbols represent the individual experiments; the dashed line, the mean response; and the dotted line within the violin, the median. (**B2**) The somatic input resistance was determined by adjusting a linear regression. (**B3**,**B4**) Somatic membrane time constant and capacitance measurements. No statistical differences were found among the experimental groups. (**C**) Scatter plot; each symbol represents the average of four consecutive traces obtained at 0.067 Hz. The responses were stable up to 20 min after the beginning of the recording. Upper panel, representative examples of the fEPSPs acquired at the beginning (indicated as “(1)”) and end (indicated as “(2)”) of the recording. (**D**) Cumulative probability chart showing the data dispersion during the acquisition of the extracellular recording. Minimal changes were found in the amplitude/slope of the fEPSPs, indicating a stable recording. (**E**) Representative paired-pulse facilitation (PPF) acquired with an inter-stimulus interval (ISI) of 60 ms. The horizontal dashed line indicates the increased facilitation found in human tissue from both experimental conditions. **** *p* < 0.0001 vs. Responsive group.

**Figure 4 biomedicines-11-03237-f004:**
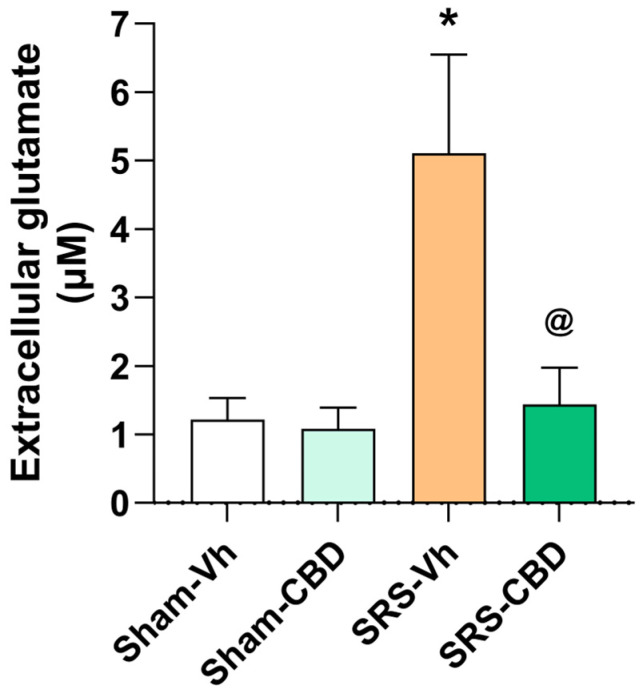
Effect of subchronic administration of CBD on the in vivo hippocampal release of glutamate in rats with epilepsy. The Sham-Vh and Sham-CBD groups showed similar glutamate concentrations. Rats from the SRE-Vh group showed higher glutamate concentration compared to the Sham groups (*p* < 0.05). In contrast, the SRS-CBD group showed a glutamate concentration similar to those of the Sham groups. Values represent mean ± standard error. One-way ANOVA followed by Tukey’s test. * *p* < 0.05 vs. Sham groups; @ *p* < 0.05 vs. SRS-Vh.

**Table 1 biomedicines-11-03237-t001:** Clinical variables of patients with drug-resistant epilepsy.

Subject	Sex	Age (Years)	Age of Seizure Onset (Years)	Duration of Epilepsy (Years)	Frequency of the Seizures (per Month)	Location of the Focus	ASMs Administered before Surgery
Drug-resistant temporal lobe epilepsy							
HUM-202	F	29	21	8	7	Right temporal lobe	LEV, VPA, CPM
HUM-203	M	29	23	6	10	Right temporal lobe	LEV
HUM-204	M	35	34	1	5	Right temporal lobe	LEV, VPA
HUM-205	F	33	1.5	31.5	2	Left temporal lobe	LEV, CBZ, TOP, GAB, VPA, LAM, LAC, ZSM
HUM-209	F	21	10	11	7	Left temporal lobe	LEV, OXC, NMP
HUM-211	M	21	13	8	12	Right temporal lobe	LEV, VPA, PAL, OXC, ZSM
HUM-220	F	23	0.5	22.5	30	Left temporal lobe	PHN, CBZ, VPA, TOP, VIG, LEV, CLZ, LAC
HUM-221	F	20	10	10	90	Left temporal lobe	PHN, CBZ, VPA, TOP, VIG, LEV, CLZ, LAC
HUM-222	M	14	3	11	5	Right temporal lobe	LEV, CBZ, OXC, CLB
HUM-223	M	16	10	6	8	Right temporal lobe	LEV, CBZ, OXC, VPA
Drug-resistant extratemporal lobe epilepsy							
HUM-206	F	11	0	11	5	Left parietal lobe	LEV, CBZ, OXC
HUM-207	F	18	5	13	3	Right frontal lobe	OXC, LEV, VPA, CLB, LAC, ZSM
HUM-208	M	1.9	0	1.9	150	Left frontal lobe	CLB, VPA, BRI, LEV, TOP, VIG
HUM-210	F	30	13	17	20	Right frontal lobe	OXC, LEV, TOP
HUM-212	M	19	4	15	7	Right frontal lobe	LEV, CLB, OXC, CBZ, LAM, VIG, LAC, TOP, BRI, LZM
HUM-213	M	18	2.5	15.5	4	Left frontal lobe	LEV, OXC, VPA, LAC, PRG
HUM-214	M	2.6	1.2	1.4	10	Right frontal lobe	LEV, LAC
HUM-217	F	7	1.25	5.75	ND	Left parietal lobe	LEV, OXC, BRI, PAL, LAC, VPA, CLB
HUM-218	F	24	16	8	ND	Right frontal lobe	LEV, LAM, VPA
HUM-219	M	4	2.75	1.25	1	Right parietal lobe	VPA

ASMs, antiseizure medications; BRI, brivaracetam; CBZ, carbamazepine; CLB, clobazam; CLZ, clorazepate; CPM, citalopram; F, female; GAB, gabapentin; LAC, lacosamide; LAM, lamotrigine; LEV, levetiracetam; LZM, lorazepam; M, male; ND, not determined; NMP, nimodipine; OXC, oxcarbazepine; PAL, perampanel; PHN, phenytoin; PRG, pregabaline; TOP, topiramate; VPA, valproic acid; VIG, vigabatrin; ZSM, zonisamide.

## Data Availability

Data are contained within the article.

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
