# Peer review of "Cannabidiol Modifies the Glutamate Over-Release in Brain Tissue of Patients and Rats with Epilepsy: A Pilot Study"

_biomedicines, 2023, doi:10.3390/biomedicines11123237_

Round 1
Reviewer 1 Report (Previous Reviewer 1)
Comments and Suggestions for Authors
The authors have improved their manuscript solving the major criticism raised in the previous revision enrolling other subjects.
Now the comparison has been performed between two groups composed of 10 subjects.
The scientific soundness is improved, despite the number being low.
I have only a minor suggestion: in the discussion section, the authors could introduce the theme of pharmacogenetics that could modulate the drug response. In this way, they may insert the following citations:
doi: 10.1016/j.neulet.2017.01.014; DOI: 10.3390/toxics9110292; doi:10.3390/ijms222111696.
Comments on the Quality of English LanguageNil.
Author Response
Thanks a lot for the comment. We have included the following sentence:
Changes in the structure of receptors and/or ion channels as consequence of genetic factors represent an explanation to the lack of effect to CBD in the brain tissue of some patients with DRE. This hypothesis highlights the relevance of pharmacogenetics studies in patients with DRE, especially in those non-responsive to CBD [51–53].
We included the references suggested by the reviewer.
Reviewer 2 Report (Previous Reviewer 2)
Comments and Suggestions for Authors
Almost all suggested issues were addressed, so the manuscript can be accepted. In the proof stage, some small mistakes can be corrected.
Author Response
We appreciate the reviewer´s comments.
Reviewer 3 Report (Previous Reviewer 3)
Comments and Suggestions for Authors
Herewith I confirm my initial highly positive evaluation of the manuscript. Authors added experimental data from animal studies and this added more value to the paper. Methods are described in details and fit well to the research aims.
Working with surgical probes from patients is very difficult and the respective findings have high worth.
Author Response
We appreciate the reviewer´s comments.
This manuscript is a resubmission of an earlier submission. The following is a list of the peer review reports and author responses from that submission.
Round 1
Reviewer 1 Report
Comments and Suggestions for Authors
This study aims to investigate if direct exposure to cannabidiol (CBD) reduces the evoked glutamate release in cortical synaptic terminals obtained from patients with Drug-resistant epilepsy (DRE).
The major criticism is related to the number of enrolled subjects: the two groups are composed of 6 and 10 subjects (DR-TLE, n=6; vs DR-ETLE, n=10). In my opinion, the number of involved subjects is too low to achieve a solid scientific conclusion. Moreover, the number is too different (6vs10-> it could be such as 60vs100, or alternatively 120vs200). Furthermore, the enrolled subjects have very different conditions (Duration of epilepsy -years; drug treatment; age)
Based on these comments, this study could be considered a pilot study, but it requires that the authors enroll the same number of subjects (10 vs. 10) in order to compare correctly their data.
Comments on the Quality of English LanguageEnglish should be improved
Author Response
The major criticism is related to the number of enrolled subjects: the two groups are composed of 6 and 10 subjects (DR-TLE, n=6; vs DR-ETLE, n=10). In my opinion, the number of involved subjects is too low to achieve a solid scientific conclusion. Moreover, the number is too different (6vs10-> it could be such as 60vs100, or alternatively 120vs200). Furthermore, the enrolled subjects have very different conditions (Duration of epilepsy -years; drug treatment; age)
Based on these comments, this study could be considered a pilot study, but it requires that the authors enroll the same number of subjects (10 vs. 10) in order to compare correctly their data.
Response: We appreciate the reviewer´s comments. We want to mention previous studies using brain tissue of patients with epilepsy at present considered with solid scientific merit. In general, these studies obtained significant results with a low number of samples.
-Hoogland et al. (2000) characterized synaptosomal GABA content using the neocortex and hippocampus of 9 patients with temporal lobe epilepsy (TLE) and 5 patients with tumor-associated epilepsy (Hoogland et al., 2000).
-Hoogland et al. (2004) evaluated the role of GABA and glutamate transporters in the extracellular concentration of these neurotransmitters using the brain tissue obtained from 6 patients with TLE (Hoogland et al., 2004).
-Rassner et al. (2016) studied the role of calcium concentration on the release of GABA from the synaptic terminals obtained from 18 patients with drug-resistant TLE and 9 with tumor-associated epilepsy.
-Cavus et al (2008) estimated the extracellular release of glutamate in the hippocampus of 17 patients with TLE.
-During and Spencer (1993) determined the release of glutamate and GABA in the hippocampus of 6 patients with TLE.
In our study, we used the tissue obtained from a total of 16 patients: 6 patients with TLE (DR-TLE group) and 10 patients with extratemporal epilepsy (DR-ETLE). This number of patients is like what has already been reported in the studies indicated above. It is important to mention that in the present study we decided to analyze the results obtained depending on the effect induced by CBD. Our results revealed that 5 samples from DR-TLE group and 7 samples from DR-ETLE group were responsive to CBD despite the different etiology of epilepsy and clinical factors. On the other hand, non-responsive tissue was also obtained from both experimental groups: 1 sample from DR-TLE group and 3 samples from DR-ETLE group. We then focused to describe the results obtained by responsive and non-responsive tissue instead of DR-TLE and DR-ETLE.
We also investigated if the ranges of the clinical data were involved in the results obtained. However, we did not find any significant correlation between the clinical data with the effects of CBD on the glutamate release. Indeed, the electrophysiology evaluation did not reveal significant differences between responsive and non-responsive tissue.
We agree with the reviewer about the low number of patients involved in the present study. However, we consider that we carried out an exhaustive evaluation of the tissue using different experimental approaches. This is very uncommon in the studies using fresh brain tissue of patients with drug-resistant epilepsy.
We reviewed the English and corrected all errors detected. Additionally, it is important to note that the manuscript was reviewed by American Manuscript Editors service before its first submission to Biomedicine. We provide a confirmation certificate.
Hoogland G, Hens JJ, De Wit M, van Veelen CW, van Huffelen AC, Gispen WH, de Graan PN. Glutamate and gamma-aminobutyric acid content and release of synaptosomes from temporal lobe epilepsy patients. J Neurosci Res. 2000 Jun 1;60(5):686-95. Doi: 10.1002/(SICI)1097-4547(20000601)60:5<686::AID-JNR14>3.0.CO;2-P.
Hoogland G, Spierenburg HA, van Veelen CW, van Rijen PC, van Huffelen AC, de Graan PN. Synaptosomal glutamate and GABA transport in patients with temporal lobe epilepsy. J Neurosci Res. 2004 Jun 15;76(6):881-90. Doi: 10.1002/jnr.20128.
Rassner MP, Moser A, Follo M, Joseph K, van Velthoven-Wurster V, Feuerstein TJ. Neocortical GABA release at high intracellular sodium and low extracellular calcium: an anti-seizure mechanism. J Neurochem. 2016 Apr;137(2):177-89. Doi: 10.1111/jnc.13555. Epub 2016 Mar 1.
Cavus I, Pan JW, Hetherington HP, Abi-Saab W, Zaveri HP, Vives KP, Krystal JH, Spencer SS, Spencer DD. Decreased hippocampal volume on MRI is associated with increased extracellular glutamate in epilepsy patients. Epilepsia. 2008 Aug;49(8):1358-66. doi: 10.1111/j.1528-1167.2008.01603.x. Epub 2008 Apr 10. PMID: 18410365.
During MJ, Spencer DD. Extracellular hippocampal glutamate and spontaneous seizure in the conscious human brain. Lancet. 1993 Jun 26;341(8861):1607-10. doi: 10.1016/0140-6736(93)90754-5. PMID: 8099987.
Reviewer 2 Report
Comments and Suggestions for Authors
The manuscript is interesting.
I suggest authors to perform only minor changes.
1) Latin expressions should be written in italic, such as in vivo, in vitro, i.e.
2) Please define the following abbreviations in full upon the first mention: di-8-ANEPPS, ACSF, RIPA, SDS, TBS-T, OPA-NAC, EGTA, HEPES, AMPA, NMDA.
Author Response
We appreciate the reviewer's comments.
- The latin expressions were written in italics.
- We made the suggested changes by adding the description of the abbreviations in their first appearance in the text.
Reviewer 3 Report
Comments and Suggestions for Authors
The manuscript describes the influence of CBD on glutamate release in isolated from epilepsy patients synaptosomes. The study is well planned and precisely performed. Description of the experimental methods is excellent and could be used as handbook with technical protocols. The number of probes is relatively low, but the complexity and the high degree of difficultness of the methods applied may justify the low number of clinical tissue samples.
I am strongly recommending the publication of the manuscript in its present form with the recommended minor correction as indicated below.
I have only one minor revision recommendation:
r. 191 "Nusschloc" should be corrected in Nussloch
Author Response
We appreciate the reviewer's comments.
Nusschloc" was changed by Nussloch.
Round 2
Reviewer 1 Report
Comments and Suggestions for Authors
The authors' response did not solve the raised criticisms. The fact that previous OLD publications (1993,2000, 2004) were published with the same number of enrolled subjects, doesn't change the raised criticisms (see the previous comments).
Comments on the Quality of English LanguageNo comments.